# Synthetic Adrenocorticotropic Peptides Modulate the Expression Pattern of Immune Genes in Rat Brain following the Early Post-Stroke Period

**DOI:** 10.3390/genes14071382

**Published:** 2023-06-30

**Authors:** Ivan B. Filippenkov, Julia A. Remizova, Vasily V. Stavchansky, Alina E. Denisova, Leonid V. Gubsky, Nikolay F. Myasoedov, Svetlana A. Limborska, Lyudmila V. Dergunova

**Affiliations:** 1Institute of Molecular Genetics of National Research Center “Kurchatov Institute”, Kurchatov Sq. 2, Moscow 123182, Russia; filippenkov-ib.img@yandex.ru (I.B.F.); utoshkautoshka@gmail.com (J.A.R.); bacbac@yandex.ru (V.V.S.); myasoedov-nf.img@yandex.ru (N.F.M.); dergunova-lv.img@yandex.ru (L.V.D.); 2Department of Neurology, Neurosurgery and Medical Genetics, Pirogov Russian National Research Medical University, Ostrovitianov Str. 1, Moscow 117997, Russia; dalina543@gmail.com (A.E.D.); gubskii@mail.ru (L.V.G.); 3Federal Center for the Brain and Neurotechnologies, Federal Biomedical Agency, Ostrovitianov Str. 1, Building 10, Moscow 117997, Russia

**Keywords:** ischemic stroke, RNA-Seq, ACTH(4-7)PGP, ACTH(6-9)PGP, tMCAO, gene expression, gene network

## Abstract

Ischemic stroke is an acute local decrease in cerebral blood flow due to a thrombus or embolus. Of particular importance is the study of the genetic systems that determine the mechanisms underlying the formation and maintenance of a therapeutic window (a time interval of up to 6 h after a stroke) when effective treatment can be provided. Here, we used a transient middle cerebral artery occlusion (tMCAO) model in rats to study two synthetic derivatives of adrenocorticotropic hormone (ACTH). The first was ACTH(4-7)PGP, which is known as Semax. It is actively used as a neuroprotective drug. The second was the ACTH(6-9)PGP peptide, which is elucidated as a prospective agent only. Using RNA-Seq analysis, we revealed hundreds of ischemia-related differentially expressed genes (DEGs), as well as 131 and 322 DEGs related to the first and second peptide at 4.5 h after tMCAO, respectively, in dorsolateral areas of the frontal cortex of rats. Furthermore, we showed that both Semax and ACTH(6-9)PGP can partially prevent changes in the immune- and neurosignaling-related gene expression profiles disturbed by the action of ischemia at 4.5 h after tMCAO. However, their different actions with regard to predominantly immune-related genes were also revealed. This study gives insight into how the transcriptome depends on the variation in the structure of the related peptides, and it is valuable from the standpoint of the development of measures for early post-stroke therapy.

## 1. Introduction

Ischemic stroke is a serious multifactorial disease accompanied by a pathology of the functions of higher nervous activity and severe cognitive impairment [1,2]. Recently, strategies for the use of neuroprotection to treat strokes have been actively developed. This path is designed to be safer and to reduce the number of side effects [3]. In this regard, it is important to understand the “therapeutic window”, that is, the time during which the recovery of cells that have retained their viability and function is still possible. The duration of the “therapeutic window” may vary slightly depending on the organism and the different patterns of ischemia, but for most cells, it is limited to 3–6 h [4,5,6,7,8].

It should be noted that stroke therapy is extremely limited; this is partly due to the unknown mechanisms of action of several potential drugs. It was shown that using approaches based on an analysis of the transcriptome of brain cells under the conditions of cerebral ischemia models in small laboratory animals can significantly detail the mechanisms underlying the neuroprotective action of a number of substances. Among these substances are peptides, including PACAP38 [9], Sal-like 4 peptide [10], OxA [11], and others.

Peptides based on adrenocorticotropic hormone (ACTH) are also of interest as neuroprotective agents [12,13,14,15]. In particular, natural ACTH(1-13), referred to as α-melanocyte-stimulating hormone, has a clear anti-inflammatory, neurogenic, and neuroprotective effect in ischemic stroke models [16,17]. However, on the basis of ACTH, peptides have been created that are completely free from hormonal activity. Thus, synthetic ACTH(4-7)PGP, which is known as Semax, is already used for the treatment of acute ischemic stroke as a neuroprotective drug, with a rapid therapeutic effect and no development of drug dependence or withdrawal syndrome [18]. With the use of the C-terminal tripeptide Pro-Gly-Pro (PGP), the duration of action of Semax is increased compared to the unmodified ACTH(4-7). Semax is also able to pass through the blood–brain barrier, including when injected [19]. Previously, gene expression assays were used to study the action of Semax under ischemic stroke in animal model conditions. These studies found that Semax was able to enhance the transcription of neurotrophins and their receptors under the conditions of a permanent middle cerebral artery occlusion (pMCAO) model in rats [20,21]. Moreover, the peptide had a significant effect on the expression of genes whose proteins were involved in the immune response [22,23]. Under rat transient middle cerebral artery occlusion (tMCAO) model conditions, Semax negatively regulated the genes in the inflammatory system and, conversely, upregulated the neuro-signaling-related genes [24]. Moreover, Semax was associated with a correction to the gene expression patterns that are disrupted during cerebral ischemia 24 h after tMCAO. The genes in this pattern were grouped into an inflammatory cluster and a neurotransmitter cluster [24,25]. A number of Semax effects have also been demonstrated at the level of protein expression in the ischemic brain of rats [26].

Here, we used the tMCAO model in rats to study ACTH(6-9)PGP as another analog of ACTH. This synthetic peptide is elucidated as a prospective drug only. Moreover, ACTH(6-9)PGP, like Semax, demonstrated nootropic and anxiolytic activity in animal experiments [27,28,29]. Moreover, the peptide increased the viability of cultured cortical neurons [29]. Using a model for Parkinson’s disease, it was demonstrated that the neuroprotective effects of the peptide were associated with modulation of the survival genes of NF-kB and NRF2 proteins [30]. In the present study, using RNA-Seq analysis, we revealed 131 and 322 differentially expressed genes in the dorsolateral areas of the frontal cortex of rats related to Semax and ACTH(6-9)PGP peptide 4.5 h after tMCAO, respectively. Furthermore, we showed that both peptides can partially prevent changes in the immune- and neuro-signaling-related gene expression profiles disturbed by the action of ischemia 4.5 h after tMCAO. However, Semax induced a number of immune genes and therefore formed a spectrum of effects on the genomic activity that were unlike those of ACTH(6-9)PGP. This study provides insight into how the transcriptome depends on variation in the structure of the related peptides and is valuable from the standpoint of the development of measures for early post-stroke therapy.

## 2. Materials and Methods

### 2.1. Animals

White 2-month-old male rats of the Wistar line (weight, 200–250 g) were obtained from AlCondi, Ltd. (Moscow, Russia) as previously described [31]. The animals were divided into four groups: “sham operation” (SH), “ischemia–reperfusion” (IR), “ischemia–reperfusion + Semax” (IS), and “ischemia–reperfusion + ACTH(6-9)PGP” (IA). Each experimental group included at least five animals.

### 2.2. Rat transient Middle Cerebral Artery Occlusion (tMCAO) Model

#### 2.2.1. Operation

The tMCAO model with 90 min occlusion was performed in accordance with the method of Koizumi et al. [32] under magnetic resonance imaging (MRI), as described previously [31]. The tMCAO details are described in Appendix A. The rats were decapitated 4.5 h after the tMCAO/sham operations.

#### 2.2.2. Peptide Administration

The peptides Semax and ACTH(6-9)PGP were administered intraperitoneally to the rats in the IS and IA groups 1.5 h and 2.5 h after the surgical procedure at doses of 100 µg/kg rat weight for each peptide, respectively, in accordance with the data from the literature [22,23,24,33,34,35,36]. The animals in the IR and SH groups were injected with saline at the same time.

### 2.3. Sample Collection and RNA Isolation

Ipsilateral fragments of the dorsolateral frontal cortex were obtained. Then, total RNA was isolated, and RNA integrity was checked using capillary electrophoresis (Experion, BioRad, Hercules, CA, USA), as previously described [27]. The RNA integrity number (RIN) was at least 9.0.

### 2.4. RNA-Seq

A polyA fraction of the total RNA was conducted with RNA-Seq analysis using an Illumina HiSeq 1500 instrument, as previously described [27]. At least 10 million reads (1/50 nt) were generated. The RNA-Seq analysis was performed with the participation of ZAO Genoanalytika, Moscow, Russia.

### 2.5. cDNA Synthesis and Real-Time Reverse Transcription Polymerase Chain Reaction (RT-PCR)

cDNA using oligo (dT)_18_ primers was synthesized, as previously described [27]. The PCR primers were selected using the OLIGO Primer Analysis Software version 6.31 and were synthesized by the Evrogen Joint Stock Company (Appendix A). Each cDNA sample was analyzed three times using RT-PCR, as previously described [27].

### 2.6. RNA-Seq Data Analysis

Three animals (*n* = 3) were included in each of the comparison groups (SH, IR, IS, IA) for RNA-Seq experiments. Cuffdiff/Cufflinks software was used for genes annotations, as previously described [31]. The levels of mRNA expression were measured using the Cuffdiff program, as previously described [24]. Only the genes that exhibited >1.5-fold changes in expression and had a *p*-value (*t*-test), adjusted using the Benjamini–Hochberg procedure, lower than 0.05 (*Padj* < 0.05) were considered.

### 2.7. Real-Time RT-PCR Data Analysis

The relative gene expression was calculated with the 2^−ΔΔCt^ method using Relative Expression Software Tool (REST) 2005 software (gene-quantification, Freising-Weihenstephan, Bavaria, Germany) [37,38], as previously described [27]. The reference gene *Gapdh* was used to normalize the expression of the cDNA samples. Each comparison group consisted of five animals. Significant differences were considered at *p* < 0.05.

### 2.8. Functional Analysis

The functions of the differentially expressed mRNAs (DEGs) were annotated using the Database for Annotation, Visualization, and Integrated Discovery (DAVID) (2021 Update) [39]. Only functional categories that had *Padj* < 0.05 were considered. Hierarchical cluster analysis of the DEGs was performed using Heatmapper (Wishart Research Group, University of Alberta, Edmonton, AB, Canada) [40]. A volcano plot was constructed using Microsoft Excel (Microsoft Office 2010). Cytoscape 3.8.2 software (Institute for Systems Biology, Seattle, WA, USA) [41] was used to visualize the regulatory network, as previously described [42].

### 2.9. Availability of Data and Material

The RNA-sequencing data were deposited in the Sequence Read Archive database under accession code PRJNA916856 (SAMN32510043-SAMN32510057, https://dataview.ncbi.nlm.nih.gov/object/PRJNA916856?reviewer=iake378bohcufsf2gpe4nnvlb7, accessed on 25 June 2023 (http://www.ncbi.nlm.nih.gov/bioproject/916856, accessed on 25 June 2023) [43].

## 3. Results

### 3.1. Magnetic Resonance Imaging (MRI)

Using diffusion-weighted imaging (DWI) and T2-weighted imaging (T2-WI), we determined the location of ischemic lesions in post-tMCAO animals. According to the MRI, after tMCAO, the animals under the influence of saline, Semax, and ACTH(6-9)PGP had an ischemic zone that was localized in the subcortical structures of the brain from the side of the occlusion or that had spread to the cortex (Appendix A). A typical MRI showing the ischemic foci of the ischemic damage area with subcortical localization in the brain of rats 4.5 h after tMCAO and peptide administration is shown in Appendix A.

### 3.2. RNA-Seq Analysis of the Effect of IR on the Dorsolateral Areas of the Frontal Cortex of Rats 4.5 h after tMCAO

Using RNA-Seq analysis, we assessed the effect of tMCAO on the mRNA levels of genes in the dorsolateral areas of the frontal cortex of rats 4.5 h after occlusion (IR vs. SH). We identified 1281 DEGs in IR vs. SH (Figure 1a, Appendix A) with 944 up- and 337 downregulated mRNAs. The top five, most highly upregulated genes in response to IR were *Hspa1*, *Ccl3*, *Ptges*, *Atf3*, and *Ccl4*, while the top five downregulated genes were *Edn3*, *Acsm5*, *Pdilt*, *Fam111a*, and *Cep131* (Figure 1b).

Real-time reverse transcription polymerase chain reaction (RT–PCR) analysis of the expression of six DEGs (*Il4r*, *Hspb1*, *Fos*, *Ccl3*, *Hes5*, and *Neurod6*) and two non-DEGs (*Grm5* and *Drd1*) was used to verify the RNA-Seq results. The characterization of the primers is shown in Appendix A. The real-time RT-PCR results adequately confirmed the RNA-Seq data (Appendix A).

### 3.3. RNA-Seq Analysis of the Effect of Semax and ACTH(6-9)PGP on the Effect of IR on the Dorsolateral Areas of the Frontal Cortex of Rats 4.5 h after tMCAO

In the dorsolateral areas of the frontal cortex of rats collected 3 h after Semax administration and 4.5 h after IR, we identified 131 DEGs (51 up- and 80 downregulated; Figure 1c, Appendix A) compared to those in the saline-treated rats (IS vs. IR). The top five most highly upregulated genes in IS vs. IR were *Mmp8*, *S100a9*, *Cep131*, *Ccl7*, and *Il1b*, whereas the top five most markedly downregulated genes were *Klhl14*, *Trdn*, *Cdhr1*, *Ly6g6e*, and *Tbx21* (Figure 1d).

Simultaneously, in the dorsolateral areas of the frontal cortex of the rats subjected to IR after ACTH(6-9)PGP administration, we identified 322 DEGs (86 up- and 236 downregulated) compared with those of the IR and saline-treated groups (IA vs. IR) (Figure 1e, Appendix A). The top five upregulated (*Fam111a*, *Plxnb2*, *Ifi27*, *Wnt3*, and *Abra*) and downregulated (*Nts*, *Trh*, *Spp1*, *Klhl14*, and *Ly6g6e*) genes in IA vs. IR are shown in Figure 1f.

Additionally, we identified 315 DEGs (87 up- and 228 downregulated) after ACTH(6-9)PGP administration compared with those in the Semax-treated group 4.5 h after tMCAO (IA vs. IS) (Appendix A, Appendix A). The volcano plot (Appendix A) and violin plot (Appendix A) show the distributions of the up- and downregulated genes in the IA and IS groups. The top five upregulated (*Fam111a*, *Prr15*, *Plxnb2*, *Wnt3*, and *Ifi27*) and downregulated (*Kcna2*, *Cep131*, *Spp1*, *Il1a*, and *Trh*) genes in IA and IS are shown in Appendix A.

### 3.4. Comparison of RNA-Seq Results for Different Groups

Our analysis found that both the Semax and the ACTH(6-9)PGP peptides modulated the IR-related gene expression profile. Thus, we identified 93 overlapping DEGs in the IS vs. IR and IR vs. SH pairwise comparisons (Figure 2a). Venn diagrams with only upregulated genes and only downregulated genes under both conditions are presented in Figure 2b,c, respectively. Interestingly, we did not find any DEGs that were downregulated in both cases (Figure 2c), but we found 38 DEGs, including *Ccl3*, *Socs3*, *Gadd45g*, *Hspb1*, *Cd44*, *Nfkbiz*, *Il4r*, *Apold1*, *Bcl6b*, and *Nfkbia*, that increased the expression in both conditions (Figure 2b). Figure 2d shows the top 10 overlapping genes that had the greatest fold change in IR vs. SH. Thus, the Semax treatment of the rats that were subjected to IR resulted in three groups of genes: those upregulated with both Semax and IR (*Ccl3*, *Atf3*, *Socs*, *Olr1*, and *Maff*); those upregulated with Semax but downregulated with IR (only *Cep131* genes, which encode centrosomal protein 131); and those downregulated with Semax but upregulated with IR (*Nmb*, *Nxph4*, *Wnt5a*, *Dlx2*, *Sp8*, etc.) (Figure 2b, Appendix A).

Our analysis also identified 136 overlapping DEGs using a comparison of IA vs. IR and IR vs. SH (Figure 2e, Appendix A). Venn diagrams with only upregulated genes and only downregulated genes under both conditions are presented in Figure 2f,g, respectively. Therefore, there were 38 upregulated genes including *Ovol2*, *Arl4d*, *Rln1*, *Fosb*, and *Cbr3*, in both the IA vs. IR and IR vs. SH pairwise comparisons (Figure 2f). Meanwhile, there were only four downregulated genes (the *Kcna2* gene, which encodes the potassium voltage-gated channel, the second member of the shaker-related subfamily; the *P2ry12* gene, which encodes purinergic receptor P2Y, G-protein coupled; the *Col1a1* gene, which encodes α-1 collagen I type; and the *Zfp483* gene, which encodes zinc finger protein 483) (Figure 2g). Furthermore, ACTH(6-9)PGP also initiated gene expression that both progressed and counteracted the effects of IR. The top 10 overlapping DEGs that had the greatest fold change in IR vs. SH are shown in Figure 2h.

After comparison of the RNA-Seq results for the IS vs. IR and IA vs. IR groups, we identified 62 and 253 DEGs that were unique for the Semax and ACTH(6-9)PGP actions, respectively. Nevertheless, the mRNA levels of 69 genes under IR conditions were altered after the two peptide treatments (Figure 2i). Then, 4 upregulated (Figure 2j) and 65 downregulated genes (Figure 2k), under both conditions, were identified and presented in Venn diagrams. Thus, all the DEGs present in the IS vs. IR and IA vs IR pairwise comparisons altered their expression co-directionally under both conditions. Figure 2l shows the top 10 overlapping genes with the greatest fold change in the IS vs. IR groups. Both the Semax and ACTH(6-9)PGP treatments in combination with IR increased the expression of four genes (*Npas4*, *Coq10b*, *Xkr6*, and *Gadd45g*), and decreased the expression levels of other genes including *Inpp5j*, *Klhl14*, *Trdn*, *Cdhr1*, *Ly6g6e*, and *Tbx21* (Figure 2l, Appendix A).

The results for the IR vs. SH, IS vs. IR, and IA vs. IR triple comparison are illustrated in a Venn diagram (Figure 3a). We identified 52 genes that were altered in all the conditions (Figure 3b, Appendix A). Thus, both the Semax and ACTH(6-9)PGP treatment initiated changes in the expression of 52 common genes (*Trh*, *Klhl14*, *Cdhr1*, *Trdn*, *Shisa8*, etc.) that counteracted the effects of IR. Additionally, the triple comparison including IR vs. SH, IS vs. IR, and IA vs. IR made it possible to identify genes whose expression level changed only under the influence of one of the influencing factors. Thus, the IR impact only, with no peptide administration, specifically changed the level of mRNA in 1104 DEGs (*Hspa1*, *Ptges*, *Ccl4*, *Stc2*, *Ptgir*, etc.). Moreover, in the IS vs. IR relative complement of the Venn diagram alone, there were 21 DEGs (*Mmp8*, *S100a9*, *Ccl7*, *Il1b*, *Il1a*, etc.). Their expression profile was specifically modulated with Semax but not the ACTH(6-9)PGP peptide 4.5 h after tMCAO. Concomitantly, there were 169 DEGs (*Abra*, *Prr15*, *Grp*, *Rpl22l1*, *Adcyap1*, etc.) that lay only in the IA vs. IR relative complement of the Venn diagram. Therefore, ACTH(6-9)PGP, but not Semax, specifically modulated their expression profile 4.5 h after tMCAO. The hierarchical cluster analysis of all the DEGs in the IR vs. SH, IS vs. IR, and IA vs. IR groups is illustrated in Figure 3c.

### 3.5. Functional Annotations of DEGs Altered in Different Comparison Groups

Using DAVID (2021 Update), which is a pathway-enrichment analysis system, we identified 82, 21, and 4 pathways associated with DEGs for the IR vs. SH, IS vs. IR, and IA vs. IR pairwise comparisons, respectively (Appendix A). Figure 4a shows the Venn diagram. Among the pathways identified, there were none that overlapped between the triple IR vs. SH, IS vs. IR, and IA vs. IR pairwise comparison. Concomitantly, 14 pathways overlapped between the IR vs. SH and IS vs. IR pairwise comparisons, and 3 pathways overlapped between the IR vs. SH and IA vs. IR pairwise comparisons. Interestingly, there were no overlapping pathways between IS vs. IR and IA vs. IR. The top five pathways with the most significant *Padj* in IR vs. SH included the MAPK and TNF signaling pathways (Figure 4b). Predominantly upregulated genes were associated with such pathways. It should be noted that the MAPK signaling pathway was included in the top five Semax-related pathways (Figure 4c). Moreover, predominantly upregulated genes were also associated with such a pathway in IS vs. IR. Simultaneously, many other pathways, including antigen processing and presentation and the intestinal immune network for IgA production, were associated with downregulated genes predominantly in IS vs. IR. Figure 4d shows all the pathways that were identified with *Padj* < 0.05 in the IA vs. IR pairwise comparison. ACTH(6-9)PGP predominantly decreased the expression level of the genes associated with cocaine addiction, extracellular matrix organization, the Wnt signaling pathway, and ECM-receptor interaction. The pathways were involved in neurotransmission, differentiation, proliferation, apoptosis, etc. It should be noted that three of these pathways (excluding Wnt signaling) were also identified in the IR vs. SH comparison and were associated with predominantly upregulated genes.

Moreover, using the DAVID database, we identified 12 pathways associated with the DEGs in IA vs. IS (Appendix A). The top five pathways with the most significant *Padj* in IA vs. IS included the cytokine–receptor interaction, PI3K-Akt, MAPK signaling, and other inflammatory pathways (Figure 1e). Predominantly, the downregulated genes in IA vs. IS were associated with such pathways.

### 3.6. The Search for Pathways That Reflect Common and Unique Gene Expression Effects of Semax and ACTH(6-9)PGP Peptides in the Early Hours of IR Conditions

In total, we identified 90 pathways for the DEGs that were altered in the different comparison groups (Appendix A). However, there were only 17 pathways that overlapped between any two pairwise comparisons (IR vs. SH and IS vs. IR; IR vs. SH and IA vs. IR; IS vs. IR and IA vs. IR). We noticed that these 17 pathways could be grouped into four clusters in accordance with the differential expression of the 76 genes involved in their presentation. Figure 5 shows the network between the pathways and the genes. In the scheme, the genes are represented with three rings of rectangular blocks colored according to their differential expression in the comparison groups. Each ring includes the same genes, but the color in the inner ring identifies the DEGs in IR vs. SH; the color in the central ring identifies the DEGs in IS vs. IR; and the color in the outer ring determines the DEGs in IA vs. IR. Four pathway clusters (PC1, PC2, PC3, and PC4) are grouped in the white hexagons. The lines connecting the genes and pathways indicate the participation of the protein products of the genes in the pathway functioning.

Thus, the first pathway cluster (PC1) included the eight pathways identified in IR vs. SH and IS vs. IR but not in IA vs. IR (Figure 5). Among the PC1 pathways were Th17, Th1, and Th2 cell differentiation, phagosome, antigen processing and presentation, and other pathways. Moreover, the PC1 pathways were associated with predominantly upregulated genes in IR vs. SH and downregulated genes in IS vs. IR. In total, 46 DEGs were associated with PC1. Among them, six genes (*RT1-Bb*, *RT1-Ba*, *RT1-Da*, *RT1-Db1*, *Fos*, and *Il1b*) had the highest number of connecting lines (six) with the pathways in PC1. However, there were 25 genes (e.g., *Hspb1*, *S100a9*, *Prkca*, and *Adcy8*) that each had only one connecting line with the pathways in PC1.

Moreover, the second pathway cluster (PC2) included four pathways (Epstein–Barr virus infection, Chagas disease, IL-17, and MAPK signaling pathways). The pathways in PC2 were identified in IR vs. SH and IS vs. IR but not in IA vs. IR (Figure 5). Furthermore, the PC2 pathways were associated with genes that were predominantly upregulated genes in IR vs. SH and upregulated genes in IS vs. IR. We identified 33 DEGs that were associated with PC2 in total. Among them were the *Ccl2*, *Ccl3*, *Il1a*, *Il1b*, and other genes. Interestingly, the *RT1-Bb*, *RT1-Ba*, *RT1-Da*, and *RT1-Db1* genes, as well as the *Cd74*, *C3*, and *Il1b* genes had two connecting lines with the pathways in PC2, whereas the other genes (e.g., *Nfkbia*, *Il1a*, *and Cd4*) had only one connecting line with the pathways in PC2. Concomitantly, the lattermost listed genes had seven, six, and five additional connections, respectively, with pathways from other clusters.

Then, we formed a third pathway cluster (PC3) that included two pathways (hematopoietic cell lineage and rheumatoid arthritis). The PC3 pathways were associated with upregulated genes in IR vs. SH (Figure 5). Concomitantly, these pathways were associated with 50% up- and 50% downregulated genes in IS vs. IR. In total, 14 DEGs (e.g., *RT1-Bb*, *RT1-Ba*, *RT1-Da*, *RT1-Db1*, and *Cd44*) were associated with PC3. Each of the listed genes had two connecting lines with the pathways in PC3, whereas the other genes (e.g., *Cd4*, *Il1a*, *Il1b*, and *Il4r*) had only one connecting line with the pathways in PC3.

Finally, the fourth pathway cluster (PC4) included three pathways (cocaine addiction, extracellular matrix organization, and ECM–receptor interaction) identified in IR vs. SH and IA vs. IR but not in IS vs. IR (Figure 5). Generally, 31 genes were associated with PC4 and satisfied the following condition: genes predominantly upregulated in IR vs. SH and downregulated in IA vs. IR. It should be noted that each of 31 genes had only 1 connecting line with the pathways in PC4. Meanwhile, some of them (*RT1-Da*, *RT1-Db1*, *Il1b*, and *Fos*) had up to 10 additional connections with pathways from other clusters.

It should be noted that 46 out of 76 genes were associated with pathways that lay within only one out of the four identified clusters. Thus, the differences in IR influence as well as the Semax and ACTH(6-9)PGP peptides effects 4.5 h after tMCAO, were illustrated with PC. Meanwhile, 30 out of 76 genes had not only intracluster but also intercluster connections. Therefore, 30 genes were nodes for 2 and more PCs. Specifically, the *RT1-Ba*, *RT1-Da*, and *RT1-Db1* genes had 11 connecting lines with pathways in all of the clusters PC1-4. These genes were upregulated in IR vs. SH and downregulated in both in the IS vs. IR and the IA vs. IR pairwise comparisons. They encoded proteins related to the class of the major histocompatibility complex (MHC) and could serve as common Semax and ACTH(6-9)PGP targets for immune modulation activity in the dorsolateral areas of the frontal cortex of rats 4.5 h after tMCAO.

## 4. Discussion

In the present study, the stroke- and peptide-related effects on the transcriptome 4.5 h after tMCAO in rat brain were studied. The studied time point is included in the therapeutic window period after stroke, in which effective treatment can be provided. The MRI for all the rats with saline, Semax, and ACTH(6-9)PGP treatment detected an ischemic focus in the subcortical structures of the brain after tMCAO. Concomitantly, the surrounding tissue in the frontal cortex of the ipsilateral hemisphere contained viable cells. Then, we assessed the genome-wide mRNA profile of ischemic penumbra cells in the dorsolateral region of the frontal cortex of the rats 4.5 h after tMCAO. Using RNA-Seq analysis, 1281 DEGs were identified with cut-offs of more than 1.5 (*Padj* < 0.05). They characterized the transcriptome response of the brain cells to IR 4.5 h after tMCAO. Additionally, real-time RT-PCR was used in this study. Both the RNA-Seq and RT-PCR methods showed similar values for the differential expression changes in all tested genes. Then, the analysis of the RNA-Seq results showed that many genes of inflammation and immunity were predominantly upregulated in response to IR. This is consistent with previous studies showing that inflammation mediated by microglial activation is involved in a post-IR injury [44,45,46]. Numerous other studies have also reported transcriptomic reactions associated with inflammation and the immune response of cells within an ischemic injury [47,48,49,50,51]. A study of changes in the mRNA level of inflammatory cytokines in mice showed that the TNF-a, IL-1b, IL-10, and TGF-b1 genes increased their expression levels in the first hours after tMCAO [52]. It should be noted that cerebral IR injury-related transcription factors including JUND, FOS, and EGR1 were revealed 3 h after tMCAO in mouse brains [53]. We also identified the DEGs of transcription factors (e.g., *Atf3*, *Sp8*, *Elf4*, *Tcf15*, and *Btaf1)* that were upregulated 4.5 h after tMCAO. Additionally, the gene that encoded insulin-like growth factor 1 (IGF-1) was among the DEGs 4.5 h after tMCAO. There is evidence of the IGF-1 role in stroke and cardiovascular pathologies [54,55]. Simultaneously, we identified downregulated genes for neurosignaling (e.g., *Zfp90*, *Edn3*, *Pdilt*, *Npy5r*, and *Cep131*) in IR vs. SH. Such a result indicates a reaction in the central nervous system following ischemic damage. Previously, we also observed a decrease in the expression of genes associated with these systems in the region of the subcortical structures containing the focus of ischemic damage 4.5 h after tMCAO [56].

The main goal of the current study focused on comparing gene expression patterns between animals treated with Semax and ACTH(6-9)PGP peptides. Here, we showed that the influence of both Semax and ACTH(6-9)PGP peptides significantly modified the expression profile of dozens of genes 4.5 h after occlusion in the dorsolateral region of the frontal cortex of rats. However, the Semax action was not associated with any DEGs in the ipsilateral subcortical structures of the rat brain 4.5 h after tMCAO based on the RNA-Seq data [24]. Thus, these results can indicate the distinction in the peptide-related reaction in different rat brain parts following the early post-stroke period.

Previously, using a microarray, we revealed that Semax regulated the expression of the immune response genes following pMCAO in the rat frontal cortex of a damaged left hemisphere [22,23]. Interestingly, a number of immune-related genes (e.g., *C2*, *Cd68*, *RT1-Db1*, *RT1-Ba*, *RT1-A1*, and *RT1-CE15*) were downregulated under Semax action 3 h after pMCAO [22]. Here, many DEGs (e.g., *Ly6g6e*, *Cd74*, *Sema3e*, *RT1-Ba*, *RT1-Da*, *RT1-Bb*, and *RT1-Db1*) were also associated with inflammation and the immune response and were mostly downregulated under Semax action in IS vs. IR 4.5 h after tMCAO. We found only some immune-related genes that were DEGs in both the pMCAO and the tMCAO models. Among them, the *RT1-Db1*, *RT1-Ba*, and *Cd74* genes were upregulated and the *Hspb1*, *Cd93*, *Zfp36*, and *Dusp1* genes were downregulated in both cases. Concomitantly, most of the genes were DEGs only in the tMCAO (e.g., *Il1a*, *Ccl3*, *Ccl2*, *Ccl7*, and *Il4r*) or pMCAO (e.g., *Cd68*, *Ccdc53*, *RT1-A1*, *RT1-CE15*, and *RT1-M6-2*) conditions. Observable differences in the gene expression pattern can be explained using objective variations in the conditions of different experiments (type of stroke model, occlusion time, brain area, and bioinformatics). Here, some of genes were shown to be encoded MHC class II proteins [57]. However, some of the genes encoding cytokines (e.g., *Il1b*, *Il1a*, *Ccl2*, *Ccl3*, and *Ccl7*), cell adhesion molecules (e.g., *Cd44* and *Cd247*), and other inflammatory proteins (e.g., *Bcl6b*, *Hspb1*, and *Socs3*) were upregulated under the action of Semax 4.5 h after tMCAO. Moreover, MAPK signaling was among the overlapped pathways for the IR- and Semax-related gene sets. So, predominantly upregulated genes were associated with the MAPK signaling pathway in both cases (Figure 4b,c). Thus, our results indicated a heterogenization in the immune and inflammatory response following Semax administration following the first hours of IR conditions.

It should be noted that the neuroprotective properties of Semax have been shown in clinical practice [18,58,59]. Moreover, we previously identified that Semax suppresses the mRNA level of genes encoding inflammatory mediators (*Il1b*, *Il6*, *Tnfa*, and *Cxcl2*) 24h after tMCAO in the rat brain [24,60]. The upregulation of some inflammatory genes (e.g., *Tnf*, *Il1b*, and *Il1a)*, revealed here 4.5 h after tMCAO, probably does not contradict the neuroprotective activity of Semax, but it could be the first stage in the effect of the peptide in IR conditions. Interestingly, stroke inflammation is a double-edged sword that can both hurt (foes branch) and help (friend branch) [61,62,63]. So, when inflammation occurs, it can lead to the recruitment of immune cells, the release of reactive compounds, edema, tissue damage, and, potentially, cell death during a stroke. Conversely, inflammation can promote tissue repair processes during the chronic phase of cerebral ischemia. Thus, immune system controls inflammation and recovery processes in the brain after damage [64,65,66,67,68]. Moreover, some cytokine molecules can perform the functions of both “friends” and “foes”. For example, the inflammatory cytokine IL-1 was also shown to induce global cerebral ischemia tolerance with the protection of the hippocampal CA1 neurons in gerbils [69]. Moreover, the anti-inflammatory role of IL-4 [64] is well known. It is possible that Semax accelerates the friend branch of inflammation directed to the natural regeneration of damaged cells and tissues. The expression pattern revealed can also be significant for the formation and maintenance of a therapeutic window after stroke.

The second studied ACTH(6-9)PGP peptide, as we identified here, modulated the expression of 322 genes in IR conditions when compared with that in ischemic rats injected with saline. ACTH(6-9)PGP-related DEGs encoded proteins involved in neurotransmission, differentiation, proliferation, apoptosis, and other processes. The similarities and differences between the IR influence and the Semax and ACTH(6-9)PGP peptide effects 4.5 h after tMCAO were also illustrated with four pathway clusters (PC1-4) (Figure 5). Such clusters were formed based on 17 pathways that overlapped between any of the two pairwise comparisons (IR vs. SH and IS vs. IR, IR vs. SH and IA vs. IR, or IS vs. IR and IA vs. IR) in accordance with the differential expression of the 76 genes involved in the pathway presentation. Meanwhile, some genes had not only intracluster but also intercluster connections. So, they could reflect the similarity in the actions of both peptides in the studied conditions. Interestingly, the *RT1-Ba*, *RT1-Da*, and *RT1-Db1* genes were upregulated in IR vs. SH and downregulated in both the IS vs. IR and the IA vs. IR pairwise comparisons and had many more intercluster connections in all of the clusters PC1-4. They encoded proteins related to class II of MHC and can serve as common Semax and ACTH(6-9)PGP targets for immune modulation activity in the dorsolateral areas of the frontal cortex of rats within the stroke therapeutic window 4.5 h after tMCAO.

The effects grouped into four PCs may be a reflection of the structural features of the peptides. A noticeable differential expression of the immune-related genes turned out to be the main effects of the Semax and ACTH(6-9)PGP peptides 4.5 h after tMCAO. First, both the Semax and the ACTH(6-9)PGP peptides have a PGP unit at the C-terminus of each peptide. Nonetheless, this effect might not relate to the influence of PGP only. We have previously shown that the effect of PGP injections on the expression of inflammatory cluster genes was unlike the Semax effect 24 h after tMCAO [25]. Thus, the inflammatory-related genes were DEGs after Semax but non-DEGs after PGP administration under IR conditions. It is likely that the similarity between the Semax and the ACTH(6-9)PGP action may be due more to the overlapped ACTH(6-7) (His-Phe) fragment of their structure.

The main unique effects of peptides also focus on immune-related genes (Figure 5, Appendix A). There was a heterogenization in the immune genetic response to Semax, including both the compensatory (PC1 and PC3) and enhancing (PC2 and PC3) effects of Semax on the effect of ischemia itself. A compensatory effect of ACTH(6-9)PGP was also observed (PC4). The manifestation of the Semax- and ACTH(6-9)PGP-specific effects may be due to unique sequences in their structures. Specifically, ACTH(4-5) (Met-Glu) was present only in Semax, and ACTH(8-9) (Arg-Trp) was present only in the ACTH(6-9)PGP peptide. Moreover, in addition to orthosteric binding, peptides are able to provide allosteric interactions between the patterns in peptide metabolites (synactone) and various types of receptors [70,71,72,73]. Thus, synactone, which formed during the proteolysis in each of the Semax and ACTH(6-9)PGP peptides, can be overlapped and can have differences [74,75,76]. Therefore, a common and unique spectrum of peptide-induced gene expression effects can be observed under IR conditions. It is possible that the corresponding effects will also be observed at the protein level. However, the correlation between transcriptome and proteome profiles is non-linear [77,78,79,80]. Therefore, local methods for the analysis of individual proteins may not be very informative. We believe that further integrative functional genetic and proteomic analyses in post-stroke cells will allow the establishment of specific regulatory axes based on the peptide activity. Overcoming this limitation will be valuable from the standpoint of developing measures for early post-stroke therapy in the future.

## 5. Conclusions

Thus, our data provide insight into the activity of related peptides with modulation of the transcriptome pattern following the early post-stroke period on a genome-wide scale. Immune-related genes manifested themselves most noticeably under the action of peptides as early as 4.5 h after tMCAO (within the therapeutic window interval) in rats. These results may serve as an indication that overcoming the consequences of a stroke occurs with the switching of the inflammatory response from the foes to the friend branch in the first post-stroke hours. Thus, the control of these processes may be the key to the problem of stroke, including the therapeutic window formation and the prevention of ischemic damage.

## Figures and Tables

**Figure 1 genes-14-01382-f001:**
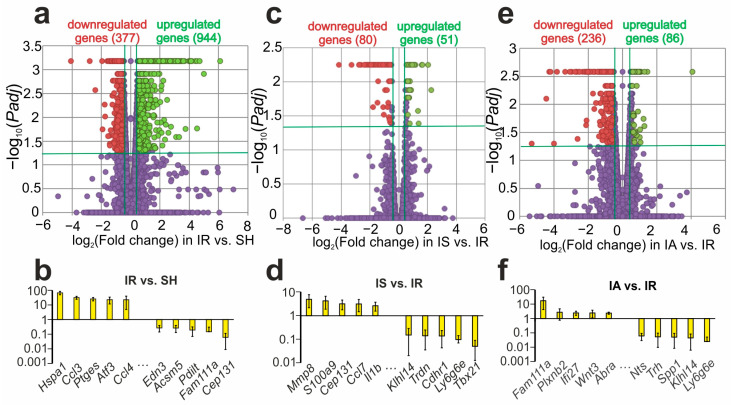
Analysis of RNA-Seq data after IR, as well as Semax or ACTH(6-9)PGP administration to the transcriptome of dorsolateral areas of the frontal cortex 4.5 h after tMCAO. (**a**,**c**,**e**) The mRNA expression changes in IR vs. SH (**a**), IS vs. IR (**c**), and IA vs. IR (**e**) groups are illustrated using volcano plots. Green, red, and purple dots depict up-, down-, and nonregulated genes, respectively. (**b,d**,**f**) The top 10 genes that exhibited the greatest fold change in expression in IR vs. SH (**b**), IS vs. IR (**d**), and IA vs. IR (**f**). The data are presented as the mean ± standard error of the mean. Only those genes with cut-off >1.5 and *Padj* < 0.05 were selected as significant results.

**Figure 2 genes-14-01382-f002:**
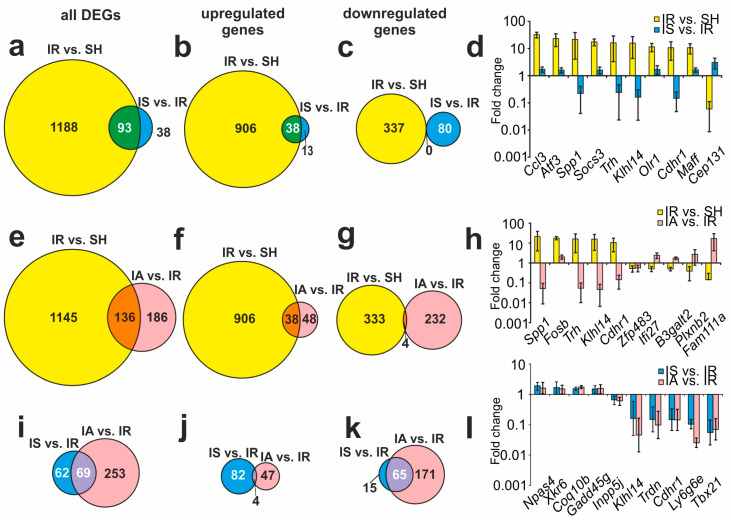
Comparison of RNA-Seq results for different groups. Venn diagrams represent results obtained in comparisons between the IR vs. SH and IS vs. IR groups (**a**–**c**); the IR vs. SH and IA vs. IR groups (**e**–**g**); and the IS vs. IR and IA vs. IR groups (**i**–**k**). All DEGs (**a**,**e**,**i**), as well as only up- (**b**,**f**,**j**) or downregulated genes (**c**,**g**,**k**) are represented in the Venn diagrams. (**d**,**h**,**l**) The top 10 genes that lie within the intersection between the gene sets in the Venn diagram (**a**,**e**,**i**), respectively, and have the greatest fold change in the IR vs. SH (**d**,**h**) and IA vs. IR (**l**) groups. The data are presented as the mean ± standard error of the mean. Only those genes with cut-off >1.5 and *Padj* < 0.05 were selected as significant results.

**Figure 3 genes-14-01382-f003:**
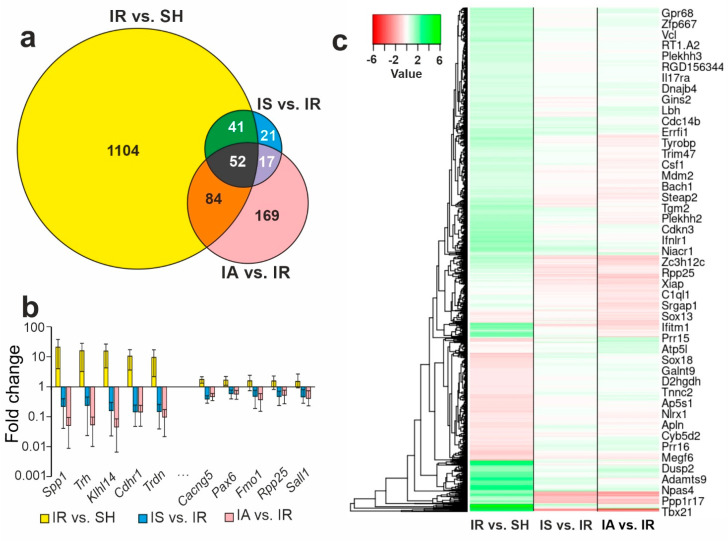
The results for the IR vs. SH, IS vs. IR, and IA vs. IR pairwise comparisons. (**a**) Venn diagrams representing comparisons between the IR vs. SH, IS vs. IR, and IA vs. IR groups. (**b**) The top 10 genes that lie within the intersection between the gene sets in the Venn diagram (**a**) and have the greatest fold change in the IR vs. SH, IS vs. IR, and IA vs. IR pairwise comparisons. The data are presented as the mean ± standard error of the mean. (**c**) DEGs in the IR vs. SH, IS vs. IR, and IA vs. IR pairwise comparisons (*n* = 3 per group) represented using heatmaps after a hierarchical cluster analysis. The comparison groups are columns and the DEGs are rows in the heatmap. Only those genes with cut-off >1.5 and Padj < 0.05 were selected as significant results.

**Figure 4 genes-14-01382-f004:**
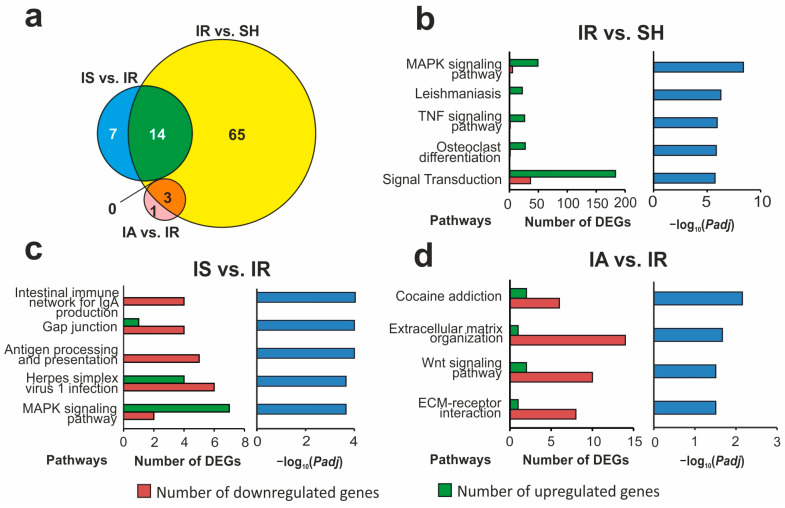
Overlapped and unique signaling pathways associated with DEGs on the dorsolateral areas of the frontal cortex of rats 4.5 h after tMCAO. The pathway enrichment analysis of DEGs in the IR vs. SH, IS vs. IR, and IA vs. IR pairwise comparisons was carried out according to DAVID (2021 Update). (**a**) Schematic comparison of DEG-related annotations obtained in the three IR vs. SH, IS vs. IR, and IA vs. IR pairwise comparisons using a Venn diagram. The number of annotations is indicated using numbers on the chart segments. (**b**–**d**) The most significant pathways among the annotations in each comparison group and the corresponding *Padj* values, as well as the number of upregulated (green) and downregulated (red) DEGs in the three pairwise comparisons IR vs. SH (**b**), IS vs. IR (**c**), and IA vs. IR (**d**) are presented. Only DEGs and pathways with *Padj* < 0.05 were selected for analysis. Each comparison group included 3 animals.

**Figure 5 genes-14-01382-f005:**
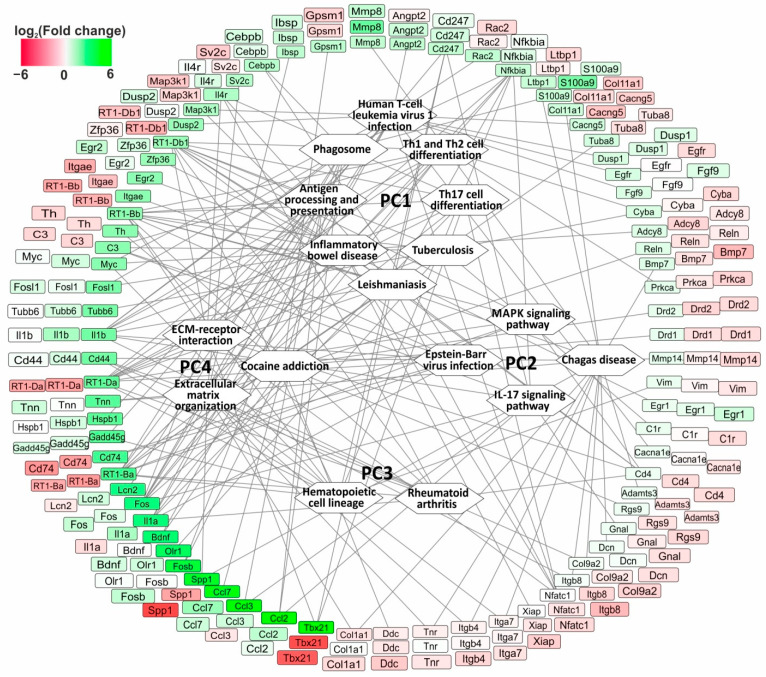
The network that reflects common and unique gene expression effects of Semax and ACTH(6-9)PGP peptides 4.5 h after tMCAO. In the scheme, genes are represented with three rings of rectangular blocks colored according to their differential expression in comparison groups. Each ring includes the same genes, but the color in the inner ring identifies DEGs in IR vs. SH, the color in the central ring identifies DEGs in the IS vs. IR, and the color in the outer ring determines the DEGs in IA vs. IR. The four pathway clusters (PC1, PC2, PC3, and PC4) are grouped in white hexagons. The lines connecting the genes and pathways indicate the participation of the protein products of genes in the pathway functioning. The DAVID database (2021 Update) was used to annotate the functions of DEGs. The network was constructed using Cytoscape 3.8.2 (Institute for Systems Biology, Seattle, WA, USA).

## Data Availability

Publicly available datasets were analyzed in this study. These data can be found here: [43].

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
