# Peer review of "Synthetic Adrenocorticotropic Peptides Modulate the Expression Pattern of Immune Genes in Rat Brain following the Early Post-Stroke Period"

_genes, 2023, doi:10.3390/genes14071382_

Round 1

Reviewer 1 Report

Authors present a very interesting paper about Synthetic Adrenocorticotropic Peptides Modulate Expression Pattern of Immune Genes in Rat Brain Following Early Post-Stroke Period.

The study aimed to determine the mechanisms for the formation and the maintenance of a therapeutic window, when the treatment can be effective. The paper is well written and important and suggest important consideration: switching the inflammatory response in first post-stroke hours.

I just suggest to consider the possible effects of ACE-inhibition on IGF-1 and IGFBP-3 concentrations, there are some evidences in cardiovascular pathologies.

Author Response

Response to the comments of Reviewer 1 to Manuscript ID: genes-2457211

Authors:

We are very grateful to the Reviewer 1 for the review and constructive comments. We carefully considered the comments of the Reviewer 1 and attached the answers to all comments.

Reviewer 1:

  1. I just suggest to consider the possible effects of ACE-inhibition on IGF-1 and IGFBP-3 concentrations, there are some evidences in cardiovascular pathologies.

Authors:

In accordance with the Reviewer’s recommendation, changes were added in the text (lines 483-485 in Mark-up copy).

Reviewer 2 Report

Filippenkov et al. used a rat model of transient middle cerebral artery occlusion (tMCAO) to investigate the effects of Semax and ACTH(6-9)PGP peptide on mRNA expression in the brain. They report in the manuscript that both peptides regulate immune-related gene expression in the frontal cortex after artery occlusion. This group of authors have shown that Semax regulates immune response gene expression using a rat model of permanent MCAO (Mol Genet Genomics. 2017; 292:635-653). Why did the authors repeat the study on Semax? Any difference in immune gene expression affected by Semax was detected in the two studies? If any, what’s the reason for the difference? Please answer and discuss these questions. The current study focuses on comparing gene expression patterns between animals treated with Semax and ACTH(6-9)PGP peptide. This seems to be the main goal of the study, which should be stressed in the manuscript.    

Other issues are as follows.

1)      The abbreviation “IS” is used for “Ischemic stroke” (line 37) and “ischemia-reperfusion + Semax” (line 94) as well. It is confusing.

2)      Do not use “upregulated DEGs” (lines 176, 188, 219, 230, 232, 245, 258, …) and “downregulated DEGs” (lines 188, 212, 219, 231, 245, 258, …). Please change them to “upregulated genes” and “downregulated genes”, respectively.

3)      The result of MRI analysis of ischemic foci at 4.5 h after tMCAO has been published previously by the very same first author (Genes. 2020;11(6):681). Therefore, subsection 3.1 and Figure 1 are not necessary and can be removed.

4)      Should “IS vs. IR” in line 250 be “IS vs. IA”?

5)      Is “much more significant (lines 296 and 311)” “the most significant”?

6)      Please number supplementary tables in the order of their citation in the text.

7)      In Supplementary Table S11, the number “0” should be added if there are no DEGs identified.

8)      In Supplementary Table S11, for IR vs. SH there are 29 upregulated genes identified for the Herpes simplex virus 1 infection pathway but there is no Padj value.

9)      In Supplementary Table S11, for IS vs. IR and IA vs. IR, the majority of DEGs have a Padj value greater than 0.05. I do not think it is necessary to put the information of these DEGs in the table.

10)   In Supplementary Table S11, 76.9% is highlighted in red, which is discordant with the footnote at the bottom of the table.

11)    Grammatical errors: “Interestingly, were no overlapped…” (line 295); “Semax action were not…” (line 426), “results can be indicate…” (line 428), “can served” (line 478), etc. English editing is required to improve the manuscript.

12)   Typos: should “inside” (line 86) be “insight” and “patter” (line 462) be “pattern”?

Numerous grammatical errors are to be fixed.

Author Response

Response to the comments of Reviewer 2 to Manuscript ID: genes-2457211

Authors:

We are very grateful to the Reviewer 2 for the review and constructive comments. We carefully considered the comments of the Reviewer 2 and attached the answers to all comments.

Reviewer 2:

  1. Filippenkov et al. used a rat model of transient middle cerebral artery occlusion (tMCAO) to investigate the effects of Semax and ACTH(6-9)PGP peptide on mRNA expression in the brain. They report in the manuscript that both peptides regulate immune-related gene expression in the frontal cortex after artery occlusion. This group of authors have shown that Semax regulates immune response gene expression using a rat model of permanent MCAO (Mol Genet Genomics. 2017; 292:635-653). Why did the authors repeat the study on Semax? Any difference in immune gene expression affected by Semax was detected in the two studies? If any, what’s the reason for the difference? Please answer and discuss these questions. The current study focuses on comparing gene expression patterns between animals treated with Semax and ACTH(6-9)PGP peptide. This seems to be the main goal of the study, which should be stressed in the manuscript.

Authors:

Indeed, the main goal of the current study focused on comparing gene expression patterns between animals treated with Semax and ACTH(6-9)PGP peptides. In accordance with the Reviewer’s recommendation, the changed were added in Discussion section of the manuscript (lines 491,492 in Mark-up copy). It should be noted that previously our team studied Semax effects using permanent left middle cerebral artery occlusion (pMCAO) model conditions. Here, we used transient (90 min) right middle cerebral artery occlusion (tMCAO) model. Thus, there are no repeats in the study on Semax. In accordance with the Reviewer’s recommendation, the comparison of the immune-related DEGs under pMCAO and tMCAO model conditions was carried out. As a result, we found only some immune-related genes that were DEGs in both the pMCAO and the tMCAO models. Among them, the RT1-Db1, RT1-Ba, and Cd74 genes were upregulated and the Hspb1, Cd93, Zfp36, and Dusp1 genes were downregulated in both cases. Concomitantly, most of the genes were DEGs only for the tMCAO (e.g., Il1a, Ccl3, Ccl2, Ccl7, and Il4r) or pMCAO (e.g., Cd68, Ccdc53, RT1-A1, RT1-CE15, and RT1-M6-2) conditions. Observable differences in the gene expression pattern can be explained by objective variations in the conditions of different experiments (type of stroke model, occlusion time, brain area, and bioinformatics).In accordance with the Reviewer’s recommendation, the changed were added in Discussion section of the manuscript (lines 503-514 in Mark-up copy).

Reviewer 2:

  • 2.1. The abbreviation “IS” is used for “Ischemic stroke” (line 37) and “ischemia-reperfusion + Semax” (line 94) as well. It is confusing

Authors:

In accordance with the Reviewer’s recommendation, changes were made (lines 39, 59, 61, 66 in Mark-up copy).

Reviewer 2:

  • 2.2. Do not use “upregulated DEGs” (lines 176, 188, 219, 230, 232, 245, 258, …) and “downregulated DEGs” (lines 188, 212, 219, 231, 245, 258, …). Please change them to “upregulated genes” and “downregulated genes”, respectively.

Authors:

In accordance with the Reviewer’s recommendation, changes were made (lines 200, 216, 236, 241, 243, 250, 251, 265, 267, 269, 284, 299, 351, 353, 356, 257, 364, 369, 403, 404, 413, 423-425, 520 in Mark-up copy, as well as in Figures 1, 2, and 4).

Reviewer 2:

  • 2.3. The result of MRI analysis of ischemic foci at 4.5 h after tMCAO has been published previously by the very same first author (Genes. 2020;11(6):681). Therefore, subsection 3.1 and Figure 1 are not necessary and can be removed.

Authors:

Previously, transcriptome response after ACTH(6-9)PGP peptide administration was not studied in tMCAO conditions. Thus, MRI analysis of ischemic foci at 4.5 h after tMCAO and ACTH(6-9)PGP action has not been published previously. In accordance with the Reviewer’s recommendation, Figure 1 was removed from the manuscript and transferred to the Supplementary files (Figure S1). Also, changes were added to subsection 3.1 of the manuscript (lines 183-193 in Mark-up copy).

Reviewer 2:

  • 2.4. Should “IS vs. IR” in line 250 be “IS vs. IA”?

We are grateful to the Reviewer 2 for the comments, but Figure 3l illustrated the overlapping genes between IS vs. IR and IA vs. IR pairwise comparisons. Top ten of them exhibited the greatest fold change in expression in IS vs. IR groups. Thus, no correction required.

Reviewer 2:

  • 2.5. Is “much more significant (lines 296 and 311)” “the most significant”?

Authors:

In accordance with the Reviewer’s recommendation, changes were made (lines 349, 367 in Mark-up copy).

Reviewer 2:

  • 2.6. Please number supplementary tables in the order of their citation in the text.

Authors:

In accordance with the Reviewer’s recommendation, changes were made (line 136, 617, in Mark-up copy).

Reviewer 2:

  • 2.7. In Supplementary Table S11, the number “0” should be added if there are no DEGs identified.

Authors:

In accordance with the Reviewer’s recommendation, changes were made in Supplementary Table S11.

Reviewer 2:

  • 2.8. In Supplementary Table S11, for IR vs. SH there are 29 upregulated genes identified for the Herpes simplex virus 1 infection pathway but there is no Padj value.

Authors:

In accordance with the Reviewer’s recommendation, changes were made in Supplementary Table S11.

Reviewer 2:

  • 2.9. In Supplementary Table S11, for IS vs. IR and IA vs. IR, the majority of DEGs have a Padj value greater than 0.05. I do not think it is necessary to put the information of these DEGs in the table.

Authors:

In accordance with the Reviewer’s recommendation, changes were made in Supplementary Table S11.

Reviewer 2:

  • 2.10. In Supplementary Table S11, 76.9% is highlighted in red, which is discordant with the footnote at the bottom of the table.

Authors:

In accordance with the Reviewer’s recommendation, changes were made in Supplementary Table S11.

Reviewer 2:

  • 2.11. Grammatical errors: “Interestingly, were no overlapped…” (line 295); “Semax action were not…” (line 426), “results can be indicate…” (line 428), “can served” (line 478), etc. English editing is required to improve the manuscript.

Authors:

In accordance with the Reviewer’s recommendation, changes were added in the text. The manuscript was undergone by English correction by MDPI English editing services: Please see order details [Manuscript ID: English-67845] MDPI English editing - Editing completed) and English-Editing-Certificate-67845 in attachment.

Reviewer 2:

  • 2.12. Typos: should “inside” (line 86) be “insight” and “patter” (line 462) be “pattern”?

Authors:

In accordance with the Reviewer’s recommendation, changes were made (lines 91, 542 in Mark-up copy). Additional changes were added throughout the text after English correction by MDPI English editing services.

Round 2

Reviewer 2 Report

All my comments have been addressed appropriately.